# Handwriting Evaluation Using Deep Learning with SensoGrip

**DOI:** 10.3390/s23115215

**Published:** 2023-05-31

**Authors:** Mugdim Bublin, Franz Werner, Andrea Kerschbaumer, Gernot Korak, Sebastian Geyer, Lena Rettinger, Erna Schönthaler, Matthias Schmid-Kietreiber

**Affiliations:** 1Computer Science and Digital Communication, Department Technics, University of Applied Sciences, FH Campus Wien, 1100 Vienna, Austria; matthias.schmid-kietreiber@fh-campuswien.ac.at; 2Health Assisting Engineering, Department Technics & Health Sciences, University of Applied Sciences, 2FH Campus Wien, 1100 Vienna, Austria; franz.werner@fh-campuswien.ac.at (F.W.); andrea.kerschbaumer@fh-campuswien.ac.at (A.K.); lena.rettinger@fh-campuswien.ac.at (L.R.); 3High Tech Manufacturing, Department Technics, University of Applied Sciences, FH Campus Wien, 1100 Vienna, Austria; gernot.korak@fh-campuswien.ac.at (G.K.); sebastian.geyer@fh-campuswien.ac.at (S.G.); 4Occupational Therapy, Department Health Sciences, University of Applied Sciences, FH Campus Wien, 1100 Vienna, Austria; erna.schoenthaler@fh-campuswien.ac.at

**Keywords:** smart sensors, machine learning, deep learning

## Abstract

Handwriting learning disabilities, such as dysgraphia, have a serious negative impact on children’s academic results, daily life and overall well-being. Early detection of dysgraphia facilitates an early start of targeted intervention. Several studies have investigated dysgraphia detection using machine learning algorithms with a digital tablet. However, these studies deployed classical machine learning algorithms with manual feature extraction and selection as well as binary classification: either dysgraphia or no dysgraphia. In this work, we investigated the fine grading of handwriting capabilities by predicting the SEMS score (between 0 and 12) with deep learning. Our approach provided a root-mean-square error of less than 1 with automatic instead of manual feature extraction and selection. Furthermore, the SensoGrip smart pen SensoGrip was used, i.e., a pen equipped with sensors to capture handwriting dynamics, instead of a tablet, enabling writing evaluation in more realistic scenarios.

## 1. Introduction

Children who experience handwriting learning disabilities may face significant negative effects on their academic success and daily life [1,2]. Cognitive, visual perceptual and fine motor skills are essential for learning to write [3]. Dysgraphia refers to the motor performance difficulties of handwriting rather than the skills of spelling or text composition. Between 5 and 34% of children never master legible, fluent and enduring handwriting despite appropriate learning over a period of 10 years [4,5]. Early detection of dysgraphia is important to initiate a targeted intervention for the improvement of the children’s writing skills, which are a basis for their academic career. Criterion-referenced assessment of handwriting skills in occupational therapy practice traditionally includes the quantitative and qualitative analysis of letter formation, spacing, alignment, legibility and handwriting speed [6].

Recently, several studies have investigated automatic dysgraphia detection using machine learning algorithms with a digital tablet [4,7,8,9,10,11,12]. Tablet sensors enable collecting signals, such as the x and y coordinates, pressure and tilt of the pen, during writing. However, these studies deployed classical machine learning algorithms, such as Random Forest4 and AdaBoost7, which require manual feature extraction and selection.

Asselborn et al. [4] extracted 53 handwriting features, which are classified into four groups: static, kinematic, pressure and tilt features. Spectrum features seem to be the most important features: six out of eight features are related to frequency. The four most discriminative features are the Bandwidth of Tremor Frequencies and the Bandwidth of Speed Frequencies (kinematic features), the Mean Speed of Pressure Change (pressure feature) and the Space Between Words (static feature). Using these 53 features and the random forest classifier, an F1-score of 97.98% was achieved. Accuracy was not reported due to an imbalanced data set (56 children with dysgraphia and 242 typically developing children). Asselborn et al. [5] used PCA to reduce the number of relevant features and to enable a more data-driven classification.

Drotar and Dobes [8] extracted 133 features using mainly the statistical measures leak min, mean, median and standard deviation of features such as velocity, acceleration, jerk, pressure, attitude, azimuth, segment/vertical/horizontal length and pen lifts. Merged from all tasks, they produced 1176 features in total. From these 1176 features, 150 features were selected using weighted k-nearest-neighbour feature selection. Finally, using these 150 features as input to the AdaBoost classifier, an accuracy of 79.5% was obtained. The data set was almost balanced: 57 children with dysgraphia and 63 normally developing children.

Dimauro et al. [9] used 13 pure text-based features, such as writing size, non-aligned left margin, skewed writing and insufficient space between words, and achieved an accuracy of 96%. Nevertheless, the data were imbalanced, with 12 out of 104 children showing dysgraphia.

In contrast, Devillaine et al. [10] used only graphical tablet sensor signals, i.e., x, y and z positions and pressure data, for feature extraction and no static features from text data. From the extracted features, 10 features were selected using linear SVM or extra trees. Different classical machine learning algorithms were tested for classification of a balanced data set consisting of 43 children with dysgraphia and 43 normally developing children. The best performance with 73.4% accuracy was achieved with the random forest algorithm. Devillaine et al. [10] also provided a comparison of different classical machine learning algorithms for dysgraphia detection, with the highest F1-score of 97.98% achieved by Asselborn et al. [4].

To summarize, all the cited papers used feature extraction and selection, which require high effort and expertise. Furthermore, the feature extraction is also subjective, i.e., based on expert experience, which might miss some important features. In this project, an LSTM deep learning algorithm was used, which automatically extracts features from raw sensor signals. To the best of our knowledge, only a few research groups have deployed deep learning for dysgraphia detection, such as Ghause et al. [11], achieving an F1-score of 98.16%. This work applied a CNN with text images as input, i.e., no dynamic features were used. Zolna et al. [12] achieved a significantly better performance in dysgraphia detection when using an RNN (more than 90%) in comparison to a CNN (25–39%). Unlike a CNN, an RNN takes into account the dynamic aspects of the writing. However, they used only the trajectory of the consecutive points as dynamic features and not pressures, velocity, accelerations and tilt as we did. They also used two-layer LSTM but with 100 neurons in each layer and a dropout layer with 50% drop probability.

In contrast to previous works, which used tablets, we used a pen equipped with sensors (SensoGrip) to capture handwriting dynamics (pressure, speed, acceleration and tilt), which looks and feels more like a real pen and enables writing evaluation in more realistic scenarios. The SensoGrip system consists of the SensoGrip pen and an app designed for the Android OS. The SensoGrip pen features an integrated microcontroller, which is able to communicate with the Android device via Bluetooth BLE. It also contains the necessary power supply, electronics and sensors for measuring tip and finger pressure, as well as an IMU MEMS three-axis accelerometer and three-axis gyroscope. The microcontroller captures the pressure data as well as the data provided by the IMU and forwards them to an app on the Android device. The user is able to acquire feedback via built-in RGB LEDs or via the mobile app.

We do not provide binary output (dysgraphia or no dysgraphia) but the scores of the German version of the Systematic Screening for Motor Handwriting difficulties (SEMS) [13]. SEMS provides values between 0 and 12 for print and a maximum of 14 points for manuscript writing. More points indicate more difficulties in writing, and therefore, these scores enable a finer granularity in handwriting evaluation. In this work, we investigated the fine grading of handwriting capabilities by predicting the SEMS score (between 0 and 12) with deep learning. We achieved a root-mean-square error (difference between the predicted SEMS score and the SEMS score estimated by expert therapists) of less than 1, with automatic instead of manual feature extraction and selection. Furthermore, we used the SensoGrip smart pen, a pen equipped with sensors to capture handwriting dynamics, instead of a tablet, enabling writing evaluation in more realistic scenarios.

In the course of this work, our results are presented and discussed, followed by a description of the applied methods.

## 2. Materials and Methods

### 2.1. The SensoGrip System

The core device for the acquirement of data was the SensoGrip pen (see Figure 1). First, we derived a comprehensive list of user story requirements using human-centred design process gathering [14]. After that, the device was developed, whose main components (1), (4) and (7) were manufactured using the additive manufacturing process of hot lithography. Apart from the outer shell (4) and the tip of the pen (7), the light guide (6), which transports the signal light from the PCB (2) to the visible LED ring, was traditionally manufactured using milling and turning PMMA stock material. The light guide was polished to make it fully transparent. The grip (3) was manufactured via indirect rapid prototyping using hot lithography to produce a split mould. Two-component silicone rubber was used to form the hollow cylindrical grip surfaces, which were then mounted on top of the corresponding finger sensors.

A NINA B306 microcontroller unit performs the processing in the pen. NINA-B3 series modules are small stand-alone Bluetooth 5 low-energy microcontroller unit (MCU) modules. For the measurement of the tip pressure, an HSFPAR004A piezoresistive sensor was used. The finger pressure was captured with an FSR 406 force-sensing resistor. The MPU5060 IMU was chosen due to low power consumption, low price, ease of use and a small form factor. It can deliver 3-axis accelerometer data as well as 3-axis gyroscope data. It also provides a built-in digital motion processor for sensor fusion. During the writing process, the different data streams provided by the sensors are captured by the SensoGrip pen and forwarded to the app on the Android device via BLE with corresponding time stamps.

Sensor data are provided in Table A1 in Appendix A.

### 2.2. Data Collection

The data collection was conducted as part of a bachelor’s thesis by two students of the Occupational Therapy Program at the University of Applied Sciences FH Campus Wien (Banhofer and Lehner [15]). The aim was to collect writing data from children without impairments as part of a qualitative and quantitative study.

For this purpose, 22 children aged 7–9 years were included in the study, 12 of which were female and 10 male.

The children used the SensoGrip pen twice during the data collection and re given the task of using the pen for at least 5 min to copy sentences from the assessment ‘SEMS’ (Vinçon, Blank and Jenetzky [16]. Expert therapists estimated SEMS scores. Both tests were conducted on the same day, with a short break in between.

According to the regulations of the ethics commission of FH Campus Wien, no ethics vote is required for such work. The development and evaluation of the test system itself were carried out prior to the aforementioned study in a comprehensive clinical trial with children in therapy in accordance with the Medical Devices Act number EK 21-042-0321. All participating children and their parents or legal guardians provided written informed consent before the study was conducted.

An overview of all tested subjects (age, gender, SEMS score) and the corresponding SEMS histogram are provided in Table A2 and Figure A2, respectively, in Appendix A.

### 2.3. Machine Learning Methods

Our model consists of a deep learning long short-term memory (LSTM) network for SEMS score prediction using time series sensor values (see Figure 2). We used LSTM networks for SEMS score prediction since these networks can cope well with vanishing gradient problems typical of training recurrent networks [17].

Mathematically speaking, we used a composition of two functions fLSTM : ℝ12×n→ℝ that predicts SEMS scores from the n values of 12 sensors, age and gender using an LSTM network. To ensure that all input variables to LSTM have uniform sequence lengths, we padded the age and gender variables (represented as a single value) with zeros, where gender was encoded as 1 for female and 0 for male.

To increase the amount of training and test data, we divided the data file for each child into 20 separate time series. In general, the time series lengths (n) may vary for each child based on their data file size. In our implementation, n usually ranges between 120 and 150 samples, but it can be further optimized. We annotated each time series with the same SEMS score, which was estimated for the whole data file.

We used an LSTM network with two hidden layers, with 70 and 50 hidden units, respectively. Between the hidden layers and after the second hidden layer, dropout layers were used to prevent overfitting (with a dropout rate of 20%). After the last hidden layer, a fully connected layer with one neuron was used, producing the regression output. For optimization, the Adam optimizer algorithm was used with an initial learning rate of 0.005 and the root-mean-square error as the performance measure. Optimization was performed in 10 iterations per epoch and 250 epochs.

We also tried several classical machine learning algorithms, such as support vector machines (SVMs), decision trees and fully connected neural networks [18], in combination with LSTM. The output of the LSTM network was combined with age and gender variables, and three discrete variables were used as input to the classical ML algorithms to obtain the final SEMS score prediction. Mathematically speaking, a composition of two functions was applied. fLSTM : ℝ10×n→ℝ predicts SEMS scores from the n values of 10 sensors using the LSTM network. fCML : ℝ3→ℝ predicts the final SEMS score using classical machine learning (CML) regression, with the SEMS score prediction from the LSTM network, age and gender as input features:SEMS_score=fCML°fLSTM=fSVM(fLSTM(X), age, gender)
where X is a matrix containing the n time series values of 10 sensors.

For model training and evaluation, we divided the data into training, validation and test sets, with ratios of 80%, 20% and 10%, respectively. The ultimate performance metrics were obtained by averaging multiple random trials on the test set. In each trial, we trained the model on the training set, identified the best model based on its performance on the validation set and assessed it on the test set. We used the standard LSTM network from MATLAB. The code snippet for building and training the LSTM network is provided in Figure A1 in Appendix A.

To make our model interpretable, we applied the methods described in [19], based on variable-wise hidden states with a mixture attention mechanism, to distinguish the contribution of each variable on SEMS score prediction (overall and for different time steps).

## 3. Results

### 3.1. Performance Measures

We used the root-mean-square error (RMSE) between the predicted and therapists’ SEMS scores as a basic measure for model evaluation:$$RMSE=\sqrt{\frac{\sum {\left(\mathrm{Predicted}\_\mathrm{SEM}-\mathrm{Therapeut}\_\mathrm{SEMS}\right)}^{2}}{Number\;of\;obeservations}}\;$$

To be able to compare our results with the results from the literature, a SEMS threshold above which dysgraphia was detected had to be defined, since in the literature, only a binary classification (dysgraphia or no dysgraphia) is used. For that purpose, we used a SEMS score at the level of 6 or above according to [13] as a possible indication for handwriting difficulties. In [13], a good agreement was achieved with SEMS score > 6 and teachers’ perceptions of handwriting difficulties (98.21% specificity and 100% sensitivity). The threshold of 6 was estimated in [13] for grade 2 children with an average age of 7.6 years, which fits well with the average age of the children (8 years) in this study.

Consequently, accuracy and the F1-score were defined using following equations:$$SEMS\;threshold\;for\;Dysgraphia\;detection:\;SEMS_{THR}=6\;$$
$$True\;positives\;\left(TP\right)=\sum \left(\left(Predicted_{SEMS}\geq SEMS_{THR}\;AND\;Therapeut_{SEMS}\geq SEMS_{THR}\right)\right)$$
$$False\;positives\;\left(FP\right)=\sum \left(\left(Predicted_{SEMS}\geq SEMS_{THR}\;AND\;Therapeut_{SEMS}<SEMS_{THR}\right)\right)\;$$
$$True\;negatives\;\left(TN\right)=\sum \left(\left(Predicted_{SEMS}\geq SEMS_{THR}\;AND\;Therapeut_{SEMS}<SEMS_{THR}\right)\right)$$
$$False\;negatives\;FN=\sum \left(\left(Predicted_{SEMS}<SEMS_{THR}\;AND\;Therapeut_{SEMS}\geq SEMS_{THR}\right)\right)$$
$$Sensitivity=\frac{TP}{TP+FN}$$
$$Specificity=\frac{TN}{TN+FP}$$
$$Precision=\frac{TP}{TP+FP}$$
$$Recall=\frac{TP}{TP+FN}$$
$$Accuracy=\frac{TP+TN}{TP+TN+FP+FN}$$
$$F1_{score}=2*Precision*\frac{Recall}{Precision+Recall}$$

### 3.2. Numerical Results and Discussion

We optimized the LSTM network by using one, two or three hidden layers and different numbers of neurons in each layer (see Table 1).

As can be seen from Table 1, the best performance was achieved with the LSTM networks with two layers. Two layers seems to be a reasonable compromise between underfitting (one-layer LSTM) and overfitting (three-layer LSTM). According to the simulations, the optimal number of neurons in the first layer was 70 and in the second layer 50. However, further optimizations are possible, since we did not simulate many different combinations of neurons per layer.

We also used a combination of classical machine learning (CML), such as support vector machines (SVMs), fully connected neural networks (FCNNs) and decision trees (DTs), with LSTM, where time series sensors values were used by LSTM for preliminary SEMS score prediction and the LSTM-predicted SEMS score was combined with discrete values of age and gender in CML to predict the final SEMS scores. However, no improvement could be achieved via LSTM alone, as shown in Table 2.

We used default hyper-parameters provided by MATLAB (e.g., SVM: epsilon = 0.2224, KernelFunction = ‘Linear’; FCNN: 1 hidden layer with 10 neurons; DT: SplitCriterioan = ‘mse’, MinParent = 10, MinLeaf = 1, Prune = ‘on’). Possibly, some improvements can be achieved by optimizing the hyper-parameters of CML algorithms. In addition, further optimization of the LSTM network alone is possible as discussed before. It is worth noting that the differences between the algorithms are relatively small. The primary goal of this study was not to determine optimal algorithm parameters but rather to demonstrate that using deep learning alone or in conjunction with classical machine learning algorithms can yield high accuracy in SEMS score prediction. One advantage of using only LSTM is that we can achieve the same or better outcome with a single algorithm instead of having to use two separate ones.

Table 3 provides performance comparisons of our model with other models from the literature. As can be seen from Table 3, we achieved the highest accuracy in comparison to published results (over 99%). We obtained a lower F1-score than the F1-score published in [4]. In this study, the standard deviation of the F1-score (more than 3%) was high, which is due to the relatively small sample size of the tested children.

In most papers, only accuracy was provided, whereas Asselborn et al. reported an F1-score since the overall accuracy might be misleading if most of the children come from one group, i.e., non-dysgraphia. The accuracy and F1-score were estimated under the assumption that the SEMS score at the level of 6 or above is an indication of handwriting difficulties, according to [13].

We achieved an RMSE value below 1. The RMSE was not reported in any of the cited papers, which merely made binary classification decisions: dysgraphia or no dysgraphia. In contrast, we performed a regression of the SEMS score, which enabled the evaluation of children’s writing capabilities on the finer granularity scale between 1 and 12 (the highest score obtained in our study was 8). Table 3 shows that if we had clinical data of children diagnosed with dysgraphia, we could attain a relatively high accuracy and F1-score by defining an appropriate SEMS threshold for dysgraphia detection, as evidenced by the high agreement between expert evaluation and our model output (indicated by the low RMSE score).

To distinguish the overall contribution of each variable on SEMS score prediction, we applied the variable-wise hidden states with a mixture attention mechanism method described in [19]. To achieve this objective, the conventional LSTM cells, which were employed in the aforementioned performance assessments, were enhanced for a hidden state matrix. In this modification, each row of the matrix contains data only from a particular input variable [19]. The code implementation was conducted according to [20], and the impact of different variables on performance is presented in Figure 3.

As can be seen from Figure 3, the importance of different sensor values for the SEMS score prediction performance was as follows (in decreasing order): angle, age, finger pressure, gender, tip pressure, gyroZ, gyroY, gyroX, accZ, accY, accX and writingSpeed.

To identify more precisely what was wrong with writing in each time step (10 ms interval in which sensor data were collected), we also applied temporal variable-wise hidden states with mixture attention according to [19]. The method in [19] explains which variable at which time step (t) has the most impact on the model output (see Figure 4; the lighter the colour, the greater the impact of the variable). This way, the children, therapist, teachers or parents can see what is wrong in each time step and possibly make appropriate corrections.

We conducted an ablation study to estimate the impact on performance when age and gender variables were excluded from the model training and prediction, as they are considered important factors in SEMS score prediction (see Table 4).

Table 4 shows that the exclusion of the gender variable resulted in an increase of more than 70% in the RMSE, while the exclusion of the age variable led to an increase of more than 50%. Moreover, when both gender and age variables were excluded, the RMSE almost doubled. This indicates that age and gender are significant factors for accurate SEMS score prediction.

We also investigated the impact of the best and worst SEMS score for handwriting on LSTM layer activations. Figure 5 shows a comparison of the input signals and layer activations in the case of the best score with a SEMS score of 0 (lefts) and the worst score with a SEMS score of 7 (rights). Again, the time stamp was 10 ms interval in which sensor data were collected, and we presented the data over 146 time stamps (one time series: 1/20 of the data file length), as defined in the Section 2.3.

As can be seen from Figure 5, the handwriting with the worse SEMS score had higher fluctuations in almost all input signals and caused higher activations of the LSTM hidden layers, especially the second and the last hidden layer.

Visualizing the decisions of our model, as presented in Figure 4 and Figure 5, facilitates the explanation of our model outputs to therapists. Thus, the trust in the model increases and diagnosis and therapy improve by cooperation between experts and AI.

## 4. Conclusions

We developed a pen with sensor capabilities called SensoGrip, which enables writing evaluation in realistic scenarios. Using deep learning (LSTM), we were able to achieve a low RMSE in SEMS score prediction (below 1). Automatic instead of manual feature extraction and selection saves time and effort. We developed and described a valuable tool for handwriting evaluation consisting of a pen equipped with sensors and deep learning software. High agreement in handwriting evaluation was achieved between the expert opinion and the output of our model (RMSE score well below 1). Additionally, the possibility of model output explanation was shown by visualizing variable importance and layer-wise neuron activations over different time steps. The advantages of our approach are high performance and a more realistic scenario (using a smart pen instead of a digital tablet).

Further data and research are needed to provide evidence of the clinical value of the tool and of the thresholds used for clinical purposes. In future works, collecting more data and optimizing the model might result in more exact handwriting evaluation and possibly dysgraphia detection. Furthermore, we will deploy our machine learning model on SensoGrip hardware to enable real-time handwriting evaluation and provide light feedback (red, yellow and green light) to children, teachers and parents. We also plan to use SensoGrip for dysgraphia diagnosis and therapy as well as for the detection and therapy of other disorders, such as like Parkinson and Alzheimer diseases.

## Figures and Tables

**Figure 1 sensors-23-05215-f001:**
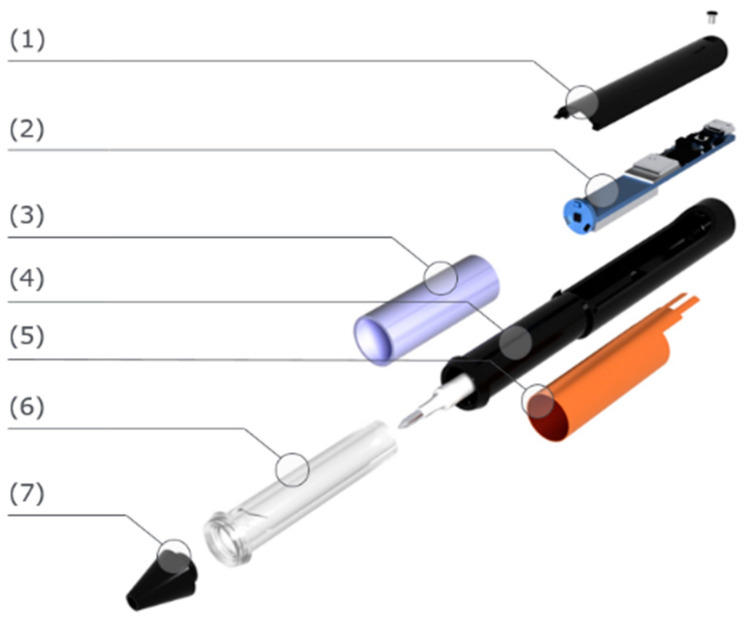
SensoGrip structure.

**Figure 2 sensors-23-05215-f002:**
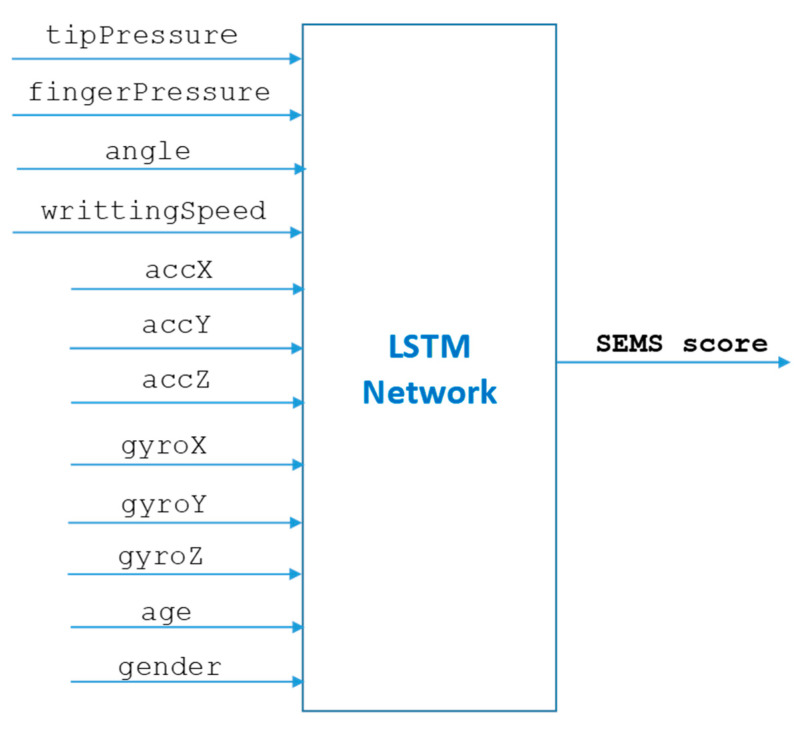
Our model for SEMS score prediction.

**Figure 3 sensors-23-05215-f003:**
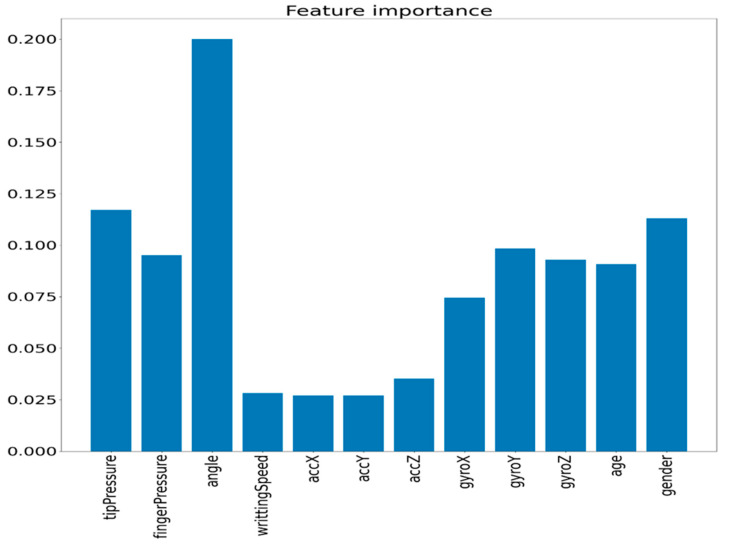
Impact of different variables (sensor values) on performance.

**Figure 4 sensors-23-05215-f004:**
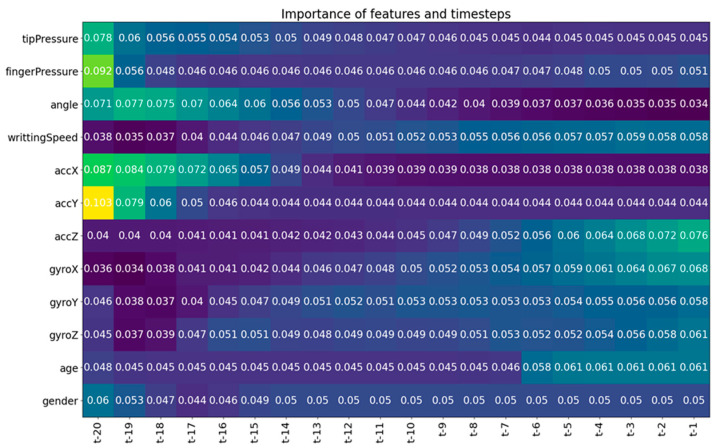
Variable importance for different time steps.

**Figure 5 sensors-23-05215-f005:**
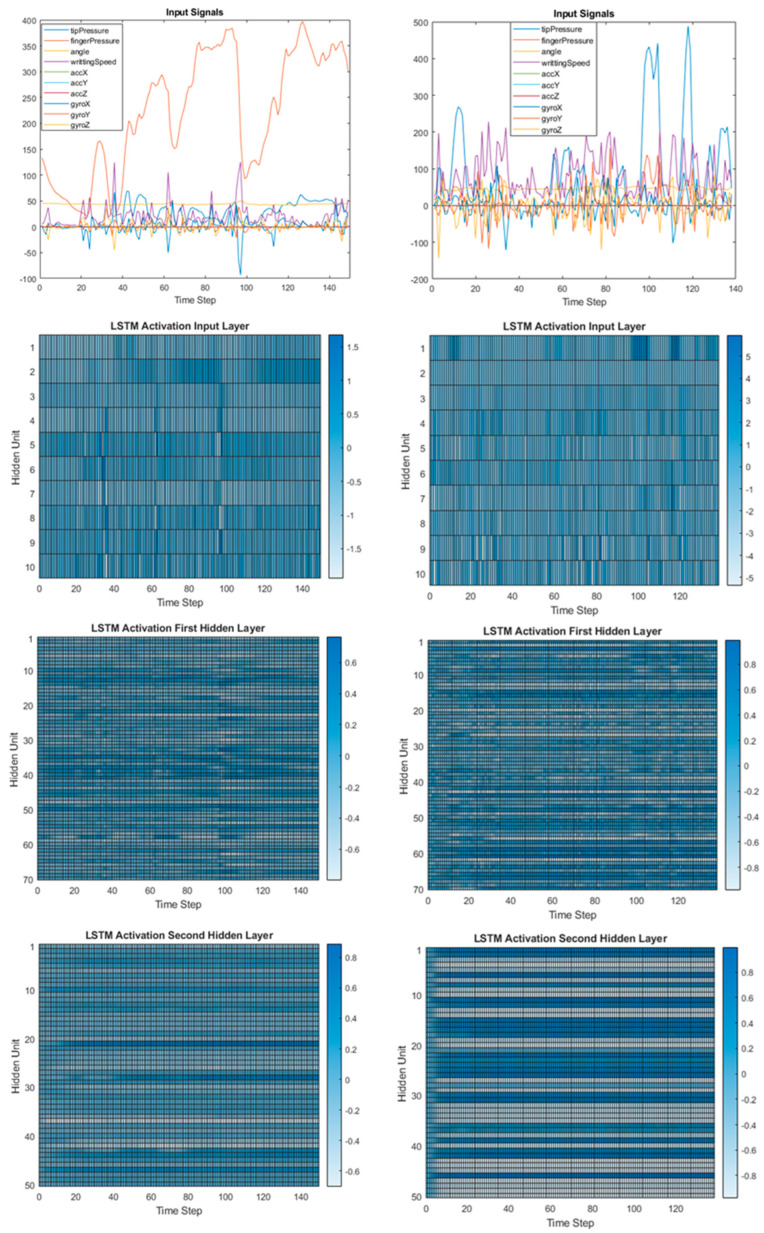
Input signals (the first row) and layers’ activations for the case of the best score (SEMS = 0; **lefts**) and the worst score (SEMS = 7; **rights**).

**Table 1 sensors-23-05215-t001:** Model optimization.

Number of Layers	Number of Hidden Units L1	Number of Hidden Units L2	Accuracy (%)	F1-Score (%)	RMSE
1	80	-	98.29	71.87	0.97
1	100	-	98.00	63.65	1.07
1	120	-	98.22	67.82	1.04
2	70	40	99.18	86.97	0.99
**2**	**70**	**50**	**99.80**	**97.78**	**0.68**
2	80	50	99.35	94.01	0.87

**Table 2 sensors-23-05215-t002:** Performance comparisons of different machine learning models.

Model	Accuracy	F1-Score	RMSE
LSTM only	**99.80**	**97.78**	**0.68**
LSTM + SVM	99.54	95.98	0.85
LSTM + FCNN	99.56	94.15	0.78
LSTM + DT	99.70	97.21	0.69

**Table 3 sensors-23-05215-t003:** Performance comparisons of our model with models from the literature.

Model	Accuracy Mean +/− Std. Dev. (%)	F1-Score Mean +/− Std (%)	RMSE Mean +/− Std. Dev.
Asselborn et al. [4]	-	97.98 +/− 2.68	-
Drotar and Dobes [8]	79.50 +/− 3	-	-
Dimauro et al. [9]	96	-	-
Devillaine et al. [10]	73.40 +/− 3.4	-	-
Ghouse et al. [11]	98.2	98.16	-
Zolna et al. [12]	90		
Our model	**99.80 +/− 0.28**	**97.78 +/− 3.14**	**0.68 +/− 0.07**

**Table 4 sensors-23-05215-t004:** Ablation study showing the impact of age and gender on RMSE performance.

Model	RMSE	RMSE Increase (%)
LSTM with all variables	0.68	0.00
LSTM without the gender variable	1.16	70.36
LSTM without the age variable	1.04	52.54
LSTM without gender and age variables	1.34	97.78

## Data Availability

The data that support the findings of this study are available from the corresponding author, M.B. upon reasonable request.

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
