# Peer review of "Handwriting Evaluation Using Deep Learning with SensoGrip"

_sensors, 2023, doi:10.3390/s23115215_

Round 1
Reviewer 1 Report
The manuscript describes the application of a very important tool for the study and the assessment of dysgraphia. The aim of the work is absolutely relevant for research on hand-writing disturbances and for clinical applications. However, in spite of the interesting statistical approach, it is not clear which are the discriminative value of the data provided by the tool. In particular, the data for the training of the network are from a small sample of 22 typically developing children aged 7-9 years, but the authors provided a threshold to detect the presence of dysgraphia. It is not clear to me how they estimated that threshold and which is the evidence that such threshold can actually discriminate the presence of dysgraphia and the severity of such disturb. If the aim of this contribute is to present to the scientific and clinical community a new interesting tool to assess the quality of handwriting, but the authors have not collected data from clinical population yet, I suggest to modify the title, in order to communicate the novelty and the promising usefulness of the tool for the assessment of handwriting disturbances, but avoiding the use of terms like "automated dysgraphia detection". Further data and research is needed to provide evidence of the clinical value of the tool and of the thresholds used for clinical purposes.
Reviewer 2 Report
This paper performs automated dysgraphia detection using deep learning + SVM. The authors collected writing data from children using a custom-made pen equipped with sensors and developed an LSTM + SVM model that evaluates their handwriting capabilities. The model predicts SEMS score, which was later used for dysgraphia classification.
Strengths:
- The paper is about an interesting topic and advances current literature by developing a smart pen that can be used in more realistic scenarios of writing disorder evaluations.
- The authors further developed a LSTM + SVM model that predicts SEMS score beyond the binary dysgraphia classification.
- Apart from a few minor mistakes, the paper is clear and easy to follow. The paper also references relevant papers.
Weaknesses and Clarifications
- The main premise of the paper is existing papers used classical machine learning (ML) which has several cons. However, the authors also use a SVM (classical ML) in their model countering the premise. To back up their premise, it would be interesting to see their results using only the deep learning model. Furthermore, age and sex look to me obvious features that significantly influence the model instead of the handwriting dynamics. At such a young age, being months apart makes a big difference in a child’s milestone as well as their writing capability. Again, it would be interesting to see the model relying on handwriting dynamics? Or detailed predictions for age groups.
- Could the authors clarify on their justification of “we do not need deep learning, since we have only three discrete variables as input”? In literature, deep learning has been applied to all sorts of data, including discreet values, text, question and answer, among others.
- How did the authors arrive to using the SVM among the classical MLs? The details and hyper parameters of the SVM aren’t also described.
- It isn’t clear which cross validation method was used or when which one was used? Was it Training, validation, test split or 10-fold? Or was the 10-fold applied on 80%, 20% or 10%?
- Did the authors have ground truth SEMS scores from a professional / therapist? It wasn’t clear to me.
- For the threshold, was score 6 or 7 used? On line 202 and 204, 6 is mentioned. However, on line 209, 7 is used.
- “To distinguish the overall contribution of each variable on SEMS score prediction, we 252 applied the variable-wise hidden states with a mixture attention mechanism method de-253 scribed in [18]. “. How was this method used on the SVM features? In 18, I believe it was applied on the LSTM only?
I look forward to the authors’ clarification and revisions.
Several grammatical errors found in the paper. Authors should revise and correct them.
